# Is a Non-Caloric Sweetener-Free Diet Good to Treat Functional Gastrointestinal Disorder Symptoms? A Randomized Controlled Trial

**DOI:** 10.3390/nu14051095

**Published:** 2022-03-05

**Authors:** Viridiana Montsserrat Mendoza-Martínez, Mónica Rocío Zavala-Solares, Aranza Jhosadara Espinosa-Flores, Karen Lorena León-Barrera, Raúl Alcántara-Suárez, José Damián Carrillo-Ruíz, Galileo Escobedo, Ernesto Roldan-Valadez, Marcela Esquivel-Velázquez, Guillermo Meléndez-Mier, Nallely Bueno-Hernández

**Affiliations:** 1Proteomics and Metabolomics Laboratory, Research Division, General Hospital of Mexico “Dr. Eduardo Liceaga”, Mexico City 06720, Mexico; viridiana_2909@hotmail.com (V.M.M.-M.); jhosadara.espinosa@gmail.com (A.J.E.-F.); karendeleonb@yahoo.com.mx (K.L.L.-B.); esquivel.marcela@gmail.com (M.E.-V.); 2School of High Studies in Health, La Salle University, Mexico City 14010, Mexico; monikazs@hotmail.com; 3Laboratory of Immunometabolism, Research Division, General Hospital of Mexico “Dr. Eduardo Liceaga”, Mexico City 06720, Mexico; raul_as02@hotmail.com (R.A.-S.); gescobedog@msn.com (G.E.); 4Neurology and Neurosurgery Unit, General Hospital of Mexico “Dr. Eduardo Liceaga”, Mexico City 06720, Mexico; josecarrilloruiz@yahoo.com; 5Faculty of Health Sciences, Mexico Anahuac University, Huixquilucan 52786, Mexico; 6Research Division, General Hospital of Mexico “Dr. Eduardo Liceaga”, Mexico City 06720, Mexico; ernest.roldan@usa.net; 7School of Public Health and Nutrition (FASPyN), Autonomous University of Nuevo Leon, Nuevo Leon 64460, Mexico

**Keywords:** non-caloric sweeteners, sucralose, functional gastrointestinal disorders, diet, irritable bowel syndrome, gastroesophageal reflux disease

## Abstract

Background: A diet containing non-caloric sweeteners (NCS) could reduce calorie intake; conversely, some animal studies suggest that NCS consumption may increase functional gastrointestinal disorder symptoms (FGDs). This study aimed to compare the effect of consuming a diet containing NCS (c-NCS) versus a non-caloric sweetener-free diet (NCS-f) on FGDs. Methods: We conducted a randomized, controlled, parallel-group study using two different diets for five weeks: the c-NCS diet contained 50–100 mg/day NCS, whereas the NCS-f diet had less than 10 mg/day NCS. At the beginning of the study (PreTx) and at the end (PostTx), we assessed FGDs, dietary intake, and NCS consumption. Results: The percentage of participants with diarrhea (PreTx = 19% vs. PstTx = 56%; *p* = 0.02), post-prandial discomfort (PreTx = 9% vs. PstTx = 39%; *p* = 0.02), constipation (PreTx = 30% vs. PostTx = 56%; *p* < 0.01), and burning (PreTx = 13% vs. PostTx = 33%; *p* < 0.01) increased in the c-NCS diet group. Conversely, abdominal pain (PreTx = 15% vs. PostTx = 3%; *p* = 0.04), post-prandial discomfort (PreTx = 26% vs. PostTx = 6%; *p* = 0.02), burning (PreTx = 15% vs. PostTx = 0%; *p* = 0.02), early satiety (PreTx = 18% vs. PostTx = 3%; *p* < 0.01), and epigastric pain (PreTx = 38% vs. PostTx = 3%; *p* < 0.01) decreased in the NCS-f diet group. Conclusion: A c-NCS diet is associated with increased FGDs, including diarrhea, post-prandial discomfort, constipation, and burning or retrosternal pain. The NCS-f diet also decreased FGDs, as well as abdominal pain, post-prandial discomfort, burning or retrosternal pain, early satiety, and epigastric pain.

## 1. Background

A diet containing non-caloric sweeteners (c-NCS) is widely used to reduce calorie intake and blood sugar peaks in overweight and obese individuals. The amount of food and beverages containing non-caloric sweeteners (NCS) has increased in the last few years. Sucralose, aspartame, acesulfame potassium (K), and saccharin are the most widely used NCS that have been approved by the United States Food and Drug Administration (FDA) [1,2].

Nowadays, many countries are experiencing transitions to a Western diet containing NCS, such as sucralose, saccharin, aspartame, acesulfame-K, and neotame [3]. In Latin America, the number of subjects consuming NCS is as high as 70%; specifically in Mexico, the most consumed NCS are sucralose (45.3%), stevia (16.7%), and mixtures of aspartame with acesulfame K (5.3%) [4,5]. Nevertheless, some studies show that NCS consumption is associated with increased functional gastrointestinal disorder symptoms (FGDs), since the gut microbiota can metabolize the unabsorbed sweeteners, leading to changes in intestinal habits. Moreover, NCS consumption is associated with altered secretion of hormones able to regulate intestinal transit, such as glucagon-like peptide-1 (GLP-1), gastric inhibitory polypeptide (GIP), peptide YY (PYY), and cholecystokinin (CCK). However, the role of NCS in gastrointestinal pathogenesis remains uncertain [6,7,8,9].

FGDs, including irritable bowel syndrome (IBS), functional dyspepsia, functional constipation, diarrhea, and gastroesophageal reflux disease (GERD), are recurrent conditions that account for at least 30% of referrals to gastroenterologists [10,11,12]. A study assessing gastrointestinal symptoms in experimental animals showed that rabbits fed 750–1000 mg/kg/day sucralose develop gastrointestinal symptoms, including perianal soiling, scouring, and cecal enlargement, compared to controls [13,14]. This evidence suggests potential deleterious actions of NCS consumption in the development of FGDs; nevertheless, there are scarce clinical studies in human populations.

This study aimed to compare the effect of consuming a c-NCS diet versus a non-caloric-sweetener-free (NCS-f) diet on FGDs in adult volunteers.

## 2. Material and Methods

### 2.1. Trial Design

We conducted a randomized, controlled, parallel-group study to evaluate the effect of NCS consumption on FGDs. We registered this trial at clinicaltrials.gov (identifier code: NCT04129762). 

### 2.2. Ethics Approval and Consent to Participate

The ethics committee of the General Hospital of Mexico “Dr. Eduardo Liceaga” approved all experimental protocols with the registration number DI/19/301/03/020 (17 October 2019). All methods were carried out following the principles of the Declaration of Helsinki and following the principles that govern research in humans in the General Health Law. All participants provided written informed consent and received an explanation of the purpose of the study and the procedures used therein. 

### 2.3. Eligibility Criteria for Participants

Inclusion criteria were women or men aged 18 to 35 years old who agreed to avoid consuming alcohol or NCS during the study. The exclusion criteria included previous diagnosis of diabetes, hypertension, and metabolic diseases. We also excluded pregnant or lactating women from the study. Participants were enlisted at the Laboratory of Proteomics and Metabolomics, Research Division at the General Hospital of Mexico Dr. Eduardo Liceaga.

### 2.4. Interventions

We randomly assigned each volunteer to one of two types of diet for five weeks: the c-NCS diet, containing 50–100 mg/day NCS (80% sucralose and 20% aspartame, acesulfame K, and saccharin), and the NCS-f diet, with <10 mg/day NCS. Volunteers in the NCS-f diet group received a list of all foods and drinks available in the market that contained NCS and a bottle of liquid sucralose (48 to 96 mg) for use in their drinks. Volunteers in the NCS-f diet had the same list and were instructed to avoid consuming items on the list for the entire study period. We registered FGDs at the beginning (PreTx) and at the end (PostTx) of the study.

### 2.5. Outcomes

NCS consumption was measured using a food frequency questionnaire (FFQ) [15] and an NCS consumption questionnaire (SCQ), which had been previously validated by our group to estimate adherence to treatment and consumption of sweeteners [16]. We also assessed FGDs according to the Rome III criteria, gastrointestinal symptoms questions [17,18,19], and the Bristol scale. The gastrointestinal symptoms questions evaluated the development of FGDs such as early satiety, constipation, diarrhea, burning or retrosternal pain, epigastric pain, abdominal pain, and post-prandial discomfort at the beginning and the end of the study. According to the Bristol scale, participants with stool consistency type 1 or 2 were diagnosed with constipation. Subjects with Bristol type 3 or 4 were supposed to have regular evacuations, and volunteers with Bristol type 5 to 7 were considered to have diarrhea. We measured body composition by electrical bioimpedance (RJL Quantum IV system, RJL Systems Inc., Clinton Township, MI, USA). A nutritionist registered anthropometric data as weight (Weighing Machine, MS 50 SECA; Hamburg, Germany), height (Scale, MD 213 SECA; Hamburg, Germany), and waist and hip circumferences (Metric Tape W606 Lufkin; CdMx Mexico). A gastroenterologist and a nutritionist with expertise in FGDs and NCS diets performed all questionnaires and measurements at the beginning (PreTx) and the end (PostTx) of the study. 

### 2.6. Sample Size and Randomization

Sample size calculation was performed with GPower v.3.1 9.2, expecting an effect size of 0.4, with an alpha error of 0.05 and a power of 80%, resulting in a sample size of 30 participants per group. Nevertheless, we increased the number of participants in the c-NCS diet group to diminish intervention bias. 

We used the website randomization.com (http://www.randomization.com, accessed on 17 September 2019) for the rationalization scheme, which we performed by blocks (6 blocks of 5 patients each). Consecutive numbers were randomized to different diets and then assigned to the patients.

### 2.7. Statistics

We evaluated the normality of data distribution using the Kolmogorov Smirnov test. We used t test, paired t test, and X^2^ test to compare intervention groups at the beginning and end of the study. In addition, we calculated mean differences (MDs) and confidence intervals (CIs) for each group (PreTx vs. PostTx values) and between groups (NCS-f vs. c-NCS). We considered a *p* value < 0.05 as statistically significant. We used the SPSS Statistics 21 program for statistical analysis.

## 3. Results

One hundred and thirty-seven participants were included in the study from October 2019 to June 2020 and randomly assigned to c-NCS or NCS-f diets. Ninety-five participants completed the study, with 34 volunteers in the NCS-f diet group and 61 volunteers in the c-NCS diet group. Figure 1 shows a flow chart illustrating the selection process of the participants according to the Consolidated Standards of Reporting Trials (CONSORT) guidelines [20].

Table 1 shows the baseline characteristics of the study population. Participants were more often women than men in both groups (59% and 62%), with a median age of 22 ± 3.2 years in the NCS-f diet group and 22 ± 3.1 years in the c-NCS diet group. There were no significant differences between groups for weight (64.9 ± 12.71 vs. 64.9 ± 12.77; *p* = 0.98), waist-to-hip ratio (0.81 ± 0.67 vs. 0.81 ± 0.65; *p* = 052), or body mass index (24.16 ± 3.8 vs. 23.9 ± 3.1; *p* = 0.75) at the beginning of the study. Consumption of carbohydrates, proteins, lipids, and calories was not different between groups at the beginning of the study. Moreover, the amount and type of consumed NCS were similar between the groups at the beginning of the study (Table 1). 

At the end of the study, in the NCS-f diet group, NCS (mg/day) consumption decreased compared to the baseline (PreTx = 42.5 ± 20.9 vs. PostTx = 2.6 ± 1.7; *p* = 0.06), whereas it tended to increase in the c-NCS diet group (PreTx = 50.03 ± 16 vs. PostTx = 74.2 ± 3.3; *p* = 0.15) without reaching significant differences. However, in the mean difference for each group, we found a considerable difference of 64.16 mg/day (95% CI, 9.47–118.85; *p* = 0.02) (Table 2).

### 3.1. Gastrointestinal Symptoms 

In the c-NCS diet group, the vast majority of FGDs increased after five weeks of dietary intervention. Specifically, the percentage of participants with diarrhea (PreTx = 19% to PostTx = 56%; *p* = 0.02), post-prandial discomfort (PreTx = 9% to PostTx = 39%; *p* = 0.02), constipation (PreTx = 30% to PostTx = 56%; *p* < 0.01), and burning or retrosternal pain (PreTx = 13% to PostTx = 33%; *p* < 0.01) significantly increased at the end of the study compared to the baseline (Figure 2A).

Participants in the NCS-f diet group showed significant decreases in all FGDs at the end of the study, especially abdominal pain (PreTx = 15% to PostTx = 3%; *p* = 0.04), post-prandial discomfort (PreTx = 26% to PostTx = 6%; *p* = 0.02), burning or retrosternal pain (PreTx = 15% to PostTx = 0%; *p* = 0.02), early satiety (PreTx = 18% to PostTx = 3%; *p* < 0.01), and epigastric pain (PreTx = 38% to PostTx = 3%; *p* < 0.01) (Figure 2B). 

### 3.2. Dietary Effects 

Weight, waist-to-hip ratio, and body mass index showed no significant changes after dietary interventions in both groups. In the NCS-f diet group, consumption of carbohydrates (PreTx = 289.4 ± 119 vs. PostTx = 219.4 ± 77; *p* = 0.001) and energy (PreTx = 2281.9 ± 1282 vs. PostTx = 1828.2 ± 1025; *p* = 0.007) significantly decreased after the intervention. Conversely, in the c-NCS diet group, consumption of carbohydrates (PreTx = 246.5 ± 108 vs. PostTx = 260.3 ± 108; *p* = 0.33), lipids (PreTx = 67.5 ± 42 vs. PostTx = 69.1 ± 42; *p* = 0.80), and energy (PreTx = 2014.3 ± 832 vs. PostTx = 2058.3 ± 832; *p* = 0.71) tended to increase without reaching significant differences; nevertheless, in terms of mean difference, the consumption of HCO significantly differed between groups (MD = 70.09; 95% CI, 9.47–118.85; *p* = 0.02) (Table 2).

Finally, volunteers from both groups exhibited a significant decrease in fat percentage after dietary interventions (NCS-f: PreTx = 35.5 ± 7.2 vs. PostTx = 34.3 ± 6.9; *p* < 0.01, and c-NCS: PreTx = 35.5 ± 5.7 vs. PostTx = 34.6 ± 6.3; *p* < 0.01). Additionally, subjects from both groups showed significant increases in fat-free mass (NCS-f: PreTx = 64.5 ± 7.2 vs. PostTx = 65.4 ± 6.9; *p* < 0.01 and c-NCS: PreTx = 64.1 ± 5.7 vs. PostTx = 65.4 ± 6.3; *p* < 0.01) and total body water (NCS-f: PreTx = 46.2 ± 5.4 vs. final = 47.6 ± 5.4; *p* < 0.01 and c-NCS: PreTx = 45.9 ± 4.5 vs. PostTx = 47.3 ± 4.9; *p* < 0.01) at the end of dietary interventions (Table 2).

## 4. Discussion

The effects of NCS on FGDs are scantly studied, and available data are predominantly from animal studies. To the best of our knowledge, this is the first study in human beings to evaluate the development of FGDs in response to a diet containing NCS, particularly sucralose. Emerging evidence suggests that FGDs occur due to changes in gut microbiota and hormones that regulate gastrointestinal motility. It is worth mentioning that despite numerous studies that have consistently shown the harmful effects of NCS on health, their consumption as part of the diet is increasing worldwide [5]. The phenomenon could be explained by the fact that many developing countries are experiencing transitions from a Mediterranean diet to a Western diet, which contains more additives in ultra-processed food and drinks (UPFDs) [2,21,22,23]. For instance, our results show that more than 60% of participants frequently consumed any NCS, an amount significantly higher than that informed in a cross-sectional study in the USA, where 41.4% of adults often reported NCS consumption [24]. It is worth mentioning that the vast majority of the study participants did not know they consumed NCS because UPFD labels did not mention the word “light”. Based on a review of ingredients mentioned in the FFQ and SCQ, several foods and drinks contained NCS. All participants consumed, on average, 42–50 mg/day NCS before study enrollment.

This prospective study demonstrates that an NCS-f diet significantly decreases the presence of FGDs in young people. We found that early satiety, burning or retrosternal pain, epigastric pain, abdominal pain, and post-prandial discomfort showed the primary effect. In fact, at the beginning of the study, more than 50% of participants had at least one FGD, which significantly decreases after five weeks of an NCS-f diet. In this group, epigastric pain (35%) and post-prandial discomfort (20%) were the leading symptoms showing evident decreases. This phenomenon could be explained by the fact that some NCS appears to cause increases in incretin secretion in the form of GLP-1, GIP, PYY, or CCK. GLP-1 decreases motility in the antro-duodeno-jejunal region and inhibits the migrating motility complex in healthy controls and IBS patients. CCK delays gastric emptying, and GIP can also slow gastric emptying [25,26]. PYY causes delayed intestinal transit [14]. Interestingly, acesulfame-K in combination with fructose might cause slow gastric emptying by increasing motilin secretion, promoting satiety just five minutes after NCS administration [27]. Thus, removing NCS from the diet may decrease incretin secretion, promoting gastrointestinal motility and gastric emptying, which could improve epigastric pain and discomfort after food or drink consumption.

Another phenomenon captured in our study is the significant increase in bowel habit alterations in the c-NCS diet group, especially diarrhea, constipation, and post-prandial discomfort. We also found a scant decrease in abdominal pain, which could relate to improvement in body composition in both groups (decrease in fat percentage and increase in fat-free mass percentage and total body water percentage). These changes could be associated with less epigastric pain. However, this could have contributed to the reduction the symptoms even in the NCS-f diet group because suboptimal body composition is associated with more gastrointestinal symptoms [28,29]. 

A hypothesis to explain the increase in FGDs in the c-NCS diet group involves changes in the intestinal environment caused by NCS consumption, especially a reduction in strictly anaerobic bacteria, inflammation promotion, and increased gastrointestinal motility, which allows for expanding pathogenic bacteria, such as *Enterobacteriaceae* and *Clostridium leptum* (Figure 3) [30,31,32,33,34].

Accumulating evidence from in vitro and in vivo studies indicated an association of sucralose ingestion with a higher amount of commensal strain subpopulations of the gut microbiota. Some of these bacteria are associated with diarrhea, constipation, or both as a consequence of changes in intestinal permeability [35,36,37,38].

Abou-Donia et al. showed that sucralose administration for 12 weeks had numerous adverse effects on the intestine of Sprague-Dawley male rats, including elevation in fecal pH that may be associated with changes in the absorption of nutrients and drugs from the gastrointestinal tract. In the same study, the authors found a decrease in beneficial anaerobic bacteria, such as *Bifidobacterium sp., Lactobacillus sp.,* and *Bacteroides sp.* [39]. It is worth mentioning that the use of *Bifidobacterium sp.* and *Lactobacillus sp.* has benefits in treating diarrhea and ameliorating IBS symptoms, such as flatulence, stool pressure, pain, and a sensation of incomplete bowel movements. The use of *Bacteroides sp.* also benefits patients by preventing digestive tract colonization by other potential pathogens [40,41,42].

Nowadays, gut microbiota modifications linked to a reduction in NCS intake are crucial; nevertheless, there have been few clinical studies, particularly in humans. One of the most important is a study by Suez et al. in 2014, wherein the authors found, among 172 individuals, a positive correlation between NCS consumption and the Enterobacteriaceae family, the Deltaproteobacteria class, and the Actinobacteria phylum [8,36,43]. Disruption of the intestinal microbiota balance by NCS may potentially interfere with numerous essential gut functions, including normal immune responses, pathogen control, gastrointestinal motility, and nutrient metabolism. 

Our results show that an NCS-f diet improves FGDs, although patients from both groups displayed a similar proportion of gastrointestinal symptoms at the beginning of the intervention. Furthermore, these changes did not depend on body composition (fat, fat-free mass, or total body water percentage), since we found no relevant clinical changes.

More studies are needed to investigate relevant doses of sweeteners in a biological context to allow for exploration of the possible molecular mechanisms through which NCS affect the gut microbiota and neuropeptide secretion, including GLP-1, GIP, PYY, CCK, and vasoactive intestinal peptide.

This study has some limitations. First of all, we enrolled a small number of patients with FGDs diagnoses. Additionally, the study population appears very young, since the prevalence of FGDs increases at the age of 30 years. An unbalanced number of male and female participants is also relevant if we consider that sex hormones may differentially affect gastrointestinal symptoms. Finally, this study did not control other dietary aspects, such as fasting periods or sugar and fiber consumption.

## 5. Conclusions

This study shows that an NCS-f diet might alleviate numerous FGDs, such as abdominal pain, post-prandial discomfort, burning or retrosternal pain, early satiety, and epigastric pain. On the contrary, a diet containing NCS appears to increase diarrhea, post-prandial discomfort, constipation, and burning or retrosternal pain. Gastroenterologists and nutritionists should take these findings into account in the treatment of FGDs. 

## Figures and Tables

**Figure 1 nutrients-14-01095-f001:**
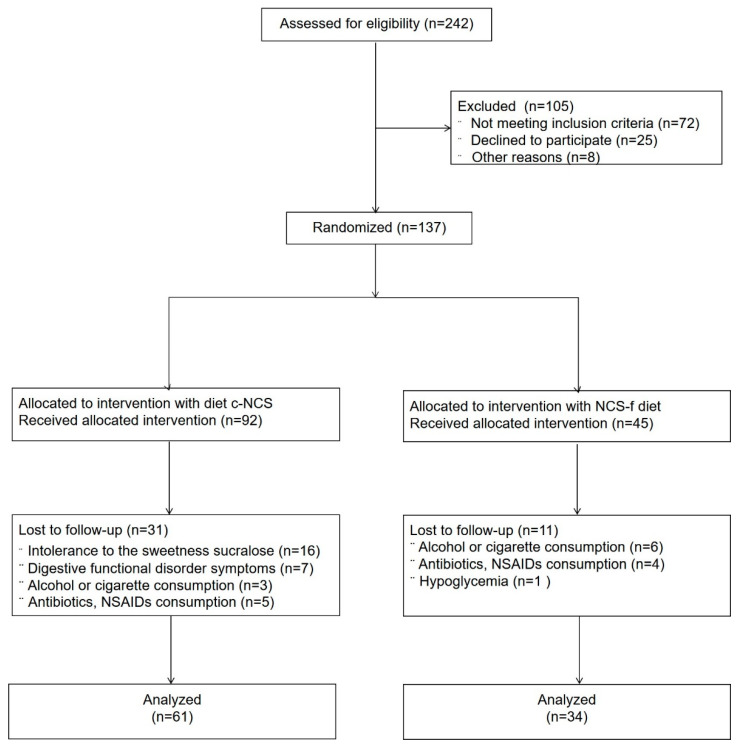
Flow chart showing the process of recruitment, randomization, follow-up, and data analysis in the study. Adherence with FGDs: functional gastrointestinal disorders symptoms, food frequency questionnaire (FFQ), NCS consumption questionnaire (SCQ), and Bristol scale. Nonsteroidal anti-inflammatory drugs (NSAIDs).

**Figure 2 nutrients-14-01095-f002:**
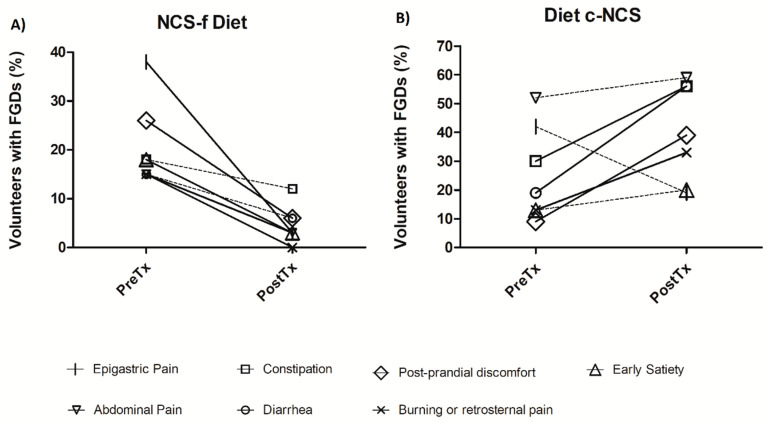
Functional gastrointestinal disorder symptoms at the beginning (PreTx) and the end (PostTx) of dietary interventions. (**A**) Represents the effect of a non-caloric sweetener-free (NCS-f) diet and (**B**) Represents the effect of a diet containing NCS (c-NCS).The solid line represents the symptoms that changed significantly (*p* < 0.05) after the 5 weeks of diet.

**Figure 3 nutrients-14-01095-f003:**
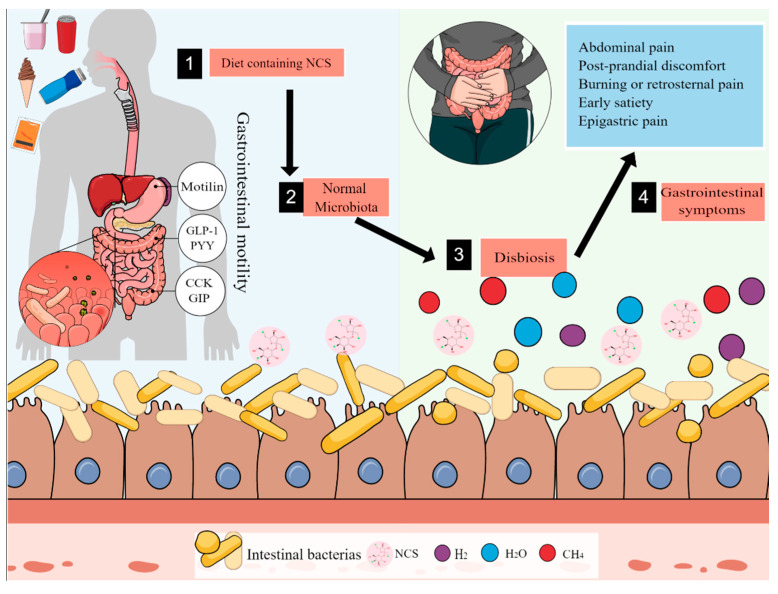
Changes in the intestinal environment and gastrointestinal motility caused by NCS consumption. Glucagon-like peptide 1 (GLP-1), gastric inhibitory polypeptide (GIP), peptide YY (PYY).

**Table 1 nutrients-14-01095-t001:** Baseline characteristics of the volunteers.

	NCS-F *N* = 34	C-NCS *N* = 61	*p **
Age, years (m ± SD)	22 ± 3.2	22 ± 3.1	0.74
Men, *n* (%)	14 (41)	23 (38)	0.73
Women, *n* (%)	20 (59)	38 (62)	0.91
Weight, kg (m ± SD)	64.9 ± 12.7	64.9 ± 12.7	0.98
WHR, (m ± SD)	0.81 ± 0.67	0.81 ± 0.65	0.52
BMI, kg/m^2^ (m ± SD)	24.16 ± 3.8	23.9 ± 3.1	0.75
Fat; % (m ± SD)	35.5 ± 7.2	35.5 ± 5.7	0.85
Fat-free mass, % (m ± SD)	64.5 ± 7.2	64.1 ± 5.7	0.85
Total body water, % (m ± SD)	46.2 ± 5.4	45.9 ± 4.5	0.79
Carbohydrates, g/day ($\bar{x}$ ± SD)	289.4 ± 118	246.6 ± 104	0.08
Protein, g/day (m ± SD)	103.9 ± 74	97.6 ± 52	0.66
Lipid, g/day (m ± SD)	71.5 ± 59	68.4 ± 43	0.78
Kilocalorie, kcal/day (m ± SD)	2264.7 ± 1247	2036.1 ± 837	0.34
Consumption of NCS, % (*n*)	23 (67)	41 (67)	0.96
Consumption amount of NCS, mg/day (m ± SE)	42.5 (20)	50.0 (16)	0.77
Adherence to treatment, %	85	90	0.26
Presence of gastrointestinal symptoms, % (*n*) **	20 (58)	33 (54)	0.65

NCS-f = Non-Caloric Sweeteners free diet, c-NCS = Containing Non-Caloric Sweeteners, PreTx = Pretreatment, PostTx = Posttreatment, m = mean, SD = standard deviation, NCS= Non-Caloric Sweeteners, WHR = waist-hip ratio, BMI = body mass index, SE = standard error. * T-Test and ** X^2^ test.

**Table 2 nutrients-14-01095-t002:** Differences and mean difference between diet groups (NCS-f and c-NCS) at the beginning (PreTx) and at the end (PostTx) of dietary interventions.

	NCS-f (*n* = 34)		Mean Difference between Pre/Post (95% CI for MD)	*p* *	c-NCS (*n* = 61)		Mean Difference between Pre/Post (95% CI for MD)	*p* *	Mean Difference between Diet Groups (95% CI for MD)	*p **
	PreTx	PostTx			PreTx	PostTx				
Weight, kg (m ± SD)	64.9 ± 12.1	64.67 ± 12.73	0.24 (−0.1; 0.6)	0.25	64.9 ± 12.7	64.9 ± 12.7	0.05 (−0.5; 0.7)	0.85	0.19 (−0.7; 1.1)	0.69
WHR, (m ± SD)	0.81 ± 0.06	0.81 ± 0.05	0.006 (−0.01; 0.02)	0.50	0.8 ± 0.6	0.8 ± 0.06	0.003 (−0.006; 0.013)	0.50	0.01 (−0.0; 0.0)	0.33
BMI, kg/m2 (m ± SD)	24.1 ± 3.8	24 ± 3.9	0.08 (−0.08; 0.24)	0.33	23.9 ± 3.1	23.9 ± 3.2	0.01 (−0.2; 0.2)	0.87	0.06 (−0.2; 0.3)	0.69
Fat, % (m ± SD)	35.5 ± 7.2	34.3 ± 6.9	0.87 (0.34; 1.41)	<0.01	35.5 ± 5.7	34.6 ± 6.3	1.1 (0.47; 1.86)	<0.01	0.56 (−6.6; 7.7)	0.87
Fat-free mass, % (m ± SD)	64.5 ± 7.2	65.4 ± 6.9	−0.87 (−1.41; −0.34)	<0.01	64.1 ± 5.7	65.4 ± 6.3	−1.1 (−1.86; −0.47)	<0.01	0.09 (−8.3; 8.4)	0.98
Total body water, % (m ± SD)	46.2 ± 5.4	47.6 ± 5.4	−1.04 (−1.6; −0.47)	<0.01	45.9 ± 4.5	47.3 ± 4.9	−1.2 (−1.87; −0.67)	<0.01	0.85 (−8.3; 10.0)	0.85
Consumption of NCS, mg/day (m ± SD)	42.5 ± 20.9	2.6 ± 1.7	39.86 (−3.1; 82.82)	0.06	50.03 ± 16	74.2 ± 3.3	−24 (16.83; 57.9)	0.15	64.16 (9.4; 118.8)	0.02
Carbohydrates, g/day (m ± SD)	289.4 ± 119	219.4 ± 77	70 (32; 107)	<0.01	246.5 ± 108	260.3 ± 108	−13.8 (−42.5; 14.9)	0.33	71.09 (18.0; 124.1)	<0.05
Protein, g/day (m ± SD)	104.7 ± 77	83.8 ± 67	20.8 (1.5; 40.1)	0.03	94.3 ± 51	87.8 ± 41	6.5 (−8.3; 21.3)	0.38	6.54 (−19.8; 32.9)	0.62
Lipid, g/day (m ± SD)	73.2 ± 61	63 ± 51	10.1 (−6.7; 27)	0.22	67.5 ± 42	69.1 ± 42	−1.6 (−14.7; 11.5)	0.80	6.18 (−15.6; 27.9)	0.57
Kilocalorie, kcal/day (m ± SD)	2281 ± 1282	1828 ± 1025	453 (132; 774)	<0.01	2014 ± 832	2058 ± 832	551 (−420; 195)	0.71	363 (−82.3; 810.0)	0.10
Presence of FGDs, % (*n*)	20 (58)	8 (23)	N.A.	0.02 **	31 (50)	39 (63)	N.A.	0.07	N.A.	<0.05 **

NCS-f = non-caloric-sweetener-free diet, c-NCS = diet containing non-caloric sweeteners, PreTx = pretreatment, PostTx = posttreatment, SD = standard deviation, NCS = non-caloric sweeteners, FGDs = functional gastrointestinal disorder symptoms, CI = confidence interval, WHR = waist–hip ratio, BMI = body mass index, m = mean, SE = standard error, MD = mean difference,* paired t test, and ** X^2^ test.

## Data Availability

We have made publicly and freely available without restriction the data described in the manuscript, at: https://www.dropbox.com/s/3riwj6oz7lprvqy/Base%20de%20Datos.sav?dl=0 (accessed on 20 December 2021).

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
