# Peer review of "Is a Non-Caloric Sweetener-Free Diet Good to Treat Functional Gastrointestinal Disorder Symptoms? A Randomized Controlled Trial"

_nutrients, 2022, doi:10.3390/nu14051095_

Round 1
Reviewer 1 Report
The study conducted by andrea et al is very interesting. It highlights a very interesting topic, the consumption of non-calorie sweeteners in age.
I suggest the authors to prepare a graphical abstrac, in this way the result is more immediate.
The authors should put more emphasis on the role of the microbiota (if they have any data).
The authors think that the quality of life is certainly better, so they should discuss this in the discussion, in spite of other diseases as well, not just intestinal
Author Response
Dear Reviewer
We have done our best to respond to your comments however, we have realized that there is probably an error in the paper that you evaluated for the following reasons.
- We do not have co-authors or references from Andrea et al.
- The paper did not evaluate the consumption of non-caloric sweeteners in age.
- The work already has a graphic abstract which we attach.
- We have previously published a paper that evaluates the sweetener's consumption with the microbiota and we have added it to the discussion and bibliography.
- We do not evaluate life quality.
We greatly appreciate your report and we are waiting for any comments or suggestions.
Reviewer 2 Report
This manuscript examined the relationship between NCS and gastrointestinal disorder symptoms. As a result, it suggests that the NCS-f diet may mitigate numerous FGDs and benefit people.
The authors have some small issues that need to be improved.
1. How many doctors and dietitians have participated in all measurements, including physical measurement?
Also, did you check the validity and no discrepancy of the results between the measurers?
2. I don't know what the yellow ellipse on the cell in Fig3 is. Is it possible to set a legend? Is it possible to ncrease the font size of "motilin"?
In addition, at the current resolution, the structural formula of NCS is difficult to see, so please reconsider whether it should be placed.
3. The reference list is not working.Everything should be remade from the beginning.
Please correct any omissions such as pages, volumes, year, doi, etc.
Author Response
Dear Reviewer
We greatly appreciate your review of our paper. My research group and I have reviewed each of your comments in detail and we provide a response below.
- A doctor and a dietitian participated in the study in order to avoid bias in the measurements, both of whom have extensive experience in research studies and clinical evaluation.
- We have modified the image with the indicated suggestions and we send the new version for your evaluation.
- All bibliographic references were rewritten, in order to avoid errors or omissions in any data.
In order to improve the writing of the introduction and methodology, we have made small changes in the wording and in the bibliography.
We will be waiting for any other comment.